

# Advection and dispersion of bedload tracers

Eric Lajeunesse[1], Olivier Devauchelle[1], and François James[2]

[1]Institut de Physique du Globe de Paris – Sorbonne Paris Cité, Équipe de Dynamique des Fuides Géologiques,
rue Jussieu, 75238 Paris cedex 05, France
[2]MAPMO UMR 7349 CNRS & Université d'Orléans, Fédération Denis Poisson FR CNRS 2964, Université d'Orléans, BP
6759, 45067 Orléans cedex 2, France

*Correspondence to:* Eric Lajeunesse (lajeunes@ipgp.fr)

**Abstract**

We use the erosion-deposition model introduced by Charru et al. (2004) to simulate numerically the evolution of a plume of
bedload tracers entrained by a steady flow. In this model, the propagation of the plume results from the stochastic exchange of
particles between the bed and the bedload layer. We find a transition between two asymptotic regimes. At early time, the tracers,
initially at rest, are progressively set into motion by the flow. During this entrainment regime, the plume, strongly skewed in
the direction of propagation, continuously accelerates while spreading non-linearly. With time, the skewness of the plume
eventually reaches a maximum value before decreasing. This marks the transition to an advection-diffusion regime in which
the plume becomes increasingly symmetrical, spreads linearly, and advances at constant velocity. We derive analytically the
expressions of the position, the variance and the skewness of the plume, and investigate their asymptotic regimes. Our model
assumes steady state. In the field, however, bedload transport is intermittent. We show that the asymptotic regimes become
insensitive to this intermittency when expressed in terms of the distance traveled by the plume. If this finding applies to the
field, it might provide an estimate for the average bedload transport rate.

## 1 Introduction

Alluvial rivers transport the sediment that makes up their bed. From a mechanical standpoint, the flow of water applies a shear
stress on the sediment particles, and entrains some of them downstream. When the shear stress is weak, the particles remain
close to the bed surface as they travel (Shields, 1936). They roll, slide and bounce over the rough bed, until they settle down
(Fernandez-Luque and Van Beek, 1976; Van Rijn, 1984; Nino and Garcia, 1994). This process is called bedload transport.

   Bedload transport is inherently random (Einstein, 1937). A turbulent burst, or a collision with an entrained grain sometime
dislodges a resting particle. The likeliness of this event depends on the specific disposition of the surrounding particles. On
average, however, the probability of entrainment is a function of macroscopic quantities such as shear stress and grain size
(Ancey et al., 2008). Once dislodged, the velocity of a particle fluctuates significantly around its average (Lajeunesse et al.,



2010a; Furbish et al., 2012b, c, a; Roseberry et al., 2012). Finally, the particle's return to rest is yet another random event. Overall, a bedload particle spends only a small fraction of its time in motion.

Altogether, the combination of these stochastic processes results in a downstream flux of particles. Fluvial geomorphologists measure this flux by collecting moving particles in traps or Helley-Smith samplers (Leopold and Emmett, 1976; Helley and

Smith, 1971). The instantaneous sediment discharge fluctuates due to the inherent randomness of bedload transport. However, averaging measurements over time yields a mean sediment flux (Liu et al., 2008).

An alternative approach to sediment flux measurements is to follow the fate of tracer particles. In November 1960, Sayre and Hubbell (1965) deposited 18 kg of radioactive sand in the North-Loup river, a sand-bed stream located in Nebraska (USA). Using a scintillator detector, they observed that the plume of radioactive sand progressively spread as it was entrained

downstream. Tracking pebbles in gravel-bed rivers reveals a similar behavior: tracers disperse as they propagates downstream (Bradley et al., 2010; Nathan Bradley and Tucker, 2012; Hassan et al., 2013; Phillips et al., 2013).

The dispersion of tracer particles results from the randomness of bedload transport. The dispersion rate, expressed as the variance of the tracers location, depends on the observation time scale (Nikora et al., 2002; Zhang et al., 2012). At long time, the variance increases linearly with time (Sayre and Hubbell, 1965; Zhang et al., 2012). This regime corresponds to

classical diffusion (Lajeunesse et al., 2013, 2017). At intermediate time, however, the variance increases non-linearly with time. This anomalous diffusion is sometimes modeled with fractional advection-dispersion equations (Schumer et al., 2009; Ganti et al., 2010; Bradley et al., 2010). Although these phenomenological models accord with observations, the transition between transient anomalous diffusion and long term classical diffusion remains elusive.

Bedload transport occurs when the shear stress exceeds a threshold set by the grain size. Most rivers fulfill this condition

only a small fraction of the time, making sediment transport highly intermittent (Phillips et al., 2013; Phillips and Jerolmack, 2014). The rate at which tracers spread thus depends not only on the inherent randomness of bedload transport, but also on the probability distribution of the river discharge (Phillips et al., 2013).

Laboratory experiments under well-controlled conditions isolate these two effects. For instance, Lajeunesse et al. (2017) tracked a plume of dyed particles in an experimental channel. Although the flow was constant in the laboratory, the tracers

still dispersed as they traveled downstream. In this case, dispersion resulted from the inherent randomness of bedload transport only. We can decompose this randomness into two components. First, the velocity fluctuations disperse the particles (Furbish et al., 2012a, c, 2016). Secondly, the random exchange of particles between the bedload layer, where particles travel, and the sediment bed, where particles are at rest, further disperses the particles (Lajeunesse et al., 2013, 2017). This effective diffusion also occurs in chromatography experiments, where a bonded phase exchanges the analyte with the flow (Van Genuchten and

Wierenga, 1976).

In a recent paper, Lajeunesse et al. (2013) used the erosion-deposition model introduced by Charru et al. (2004) to derive the equations governing the evolution of a plume of tracers. Neglecting velocity fluctuations, they found that the second dispersion process, namely the exchange of particles between the bedload layer and the sediment bed, efficiently disperses the tracers. They also observed the transition between an initial transient and classical advection-diffusion. In the present paper, we further

this investigation. Our objective is to derive formally the contribution of the advection-exchange of particles to the dispersion





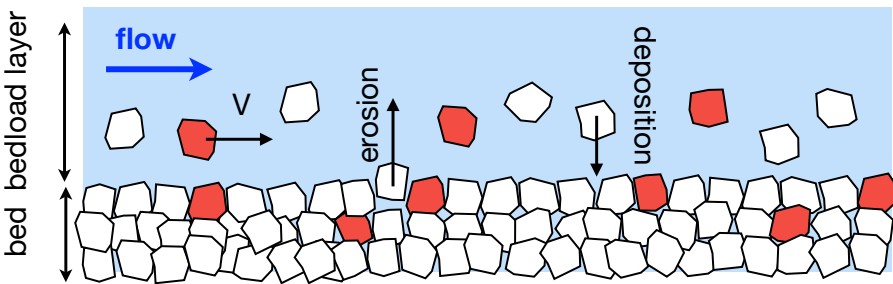

**Figure 1.** Granular bed sheared by a steady and uniform flow. The bed is a mixture of marked (red) and unmarked (white) grains.

of a plume of tracers. To do so, we briefly rederive the equations governing the evolution of a plume of tracers (Sect. 2). We simulate numerically the propagation of a plume of tracers and discuss the nature of the two asymptotic regimes evidenced in Lajeunesse et al. (2013) (Sect. 3). We analyse the long-time advection-diffusion behavior of the plume and provide an analytical expression for the diffusion coefficient and the plume velocity (Sect. 4). We derive analytically the mean, the variance and the

skewness of the tracers distribution and describe their asymptotic behavior in each regime (Sect. 5). Finally, we discuss the applicability of these results to the field (Sect. 6).

## 2    Entrainment of tracers

In most rivers, sediment are broadly distributed in size. This likely influences the dispersion of bedload tracers (Martin et al., 2012; Houssais and Lajeunesse, 2012; Pelosi et al., 2014). For the sake of simplicity, however, we restrict our analysis to a

bed of uniform particles of size $d_s$. The bed is sheared by a flow, which applies a shear stress strong enough to entrain some particles. The latter remain confined in a thin bedload layer.

For moderate values of the shear stress, the concentration of moving sediments is small, and we can neglect the interactions between particles. The erosion-deposition model introduced by Charru et al. (2004) provides an accurate description of this dilute regime, in which bedload transport is controlled by the exchange of particles between the sediment bed and the bedload

layer. This exchange sets the surface concentration of moving particles, $n_\mathrm{m}$, through mass balance:

$$\frac{\partial n_\mathrm{m}}{\partial t} + V \frac{\partial n_\mathrm{m}}{\partial x} = E - D, \tag{1}$$

where we introduce the average particle velocity $V$. $E$ is the erosion rate, defined as the number of bed particles set in motion per unit time and surface. Similarly, the deposition rate $D$ is defined as the number of bedload particles settling on the bed per unit time and surface (Charru et al., 2004; Charru, 2006; Lajeunesse et al., 2010b; Seizilles et al., 2014; Lajeunesse et al.,

2017).

To investigate the dispersion of bedload particles, we consider that some of them are marked (Fig. 1). We refer to these marked particles as "tracers", and assume that their physical properties are the same as those of unmarked particles. With these





assumptions, the mass balance for the tracers in the bedload layer reads

$$n_{\mathrm{m}} \frac{\partial \phi}{\partial t} + n_{\mathrm{m}} V \frac{\partial \phi}{\partial x} = E\psi - D\phi, \tag{2}$$

where we introduce the proportion of tracers in the moving layer, $\phi$. Similarly, $\psi$ is the proportion of tracers on the bed surface.

When subjected to varying flow and sediment discharges, the bed of a stream accumulates or releases sediments (Gintz
et al., 1996; Blom and Parker, 2004). Some particles may then be temporary buried within the bed, inducing streamwise
dispersion (Crickmore and Lean, 1962; Pelosi et al., 2014). Here, we neglect this mechanism and restrict our analysis to steady
and uniform sediment transport. Accordingly, we assume that erosion and deposition affects the bed over a depth of about
one grain diameter only. This hypothesis holds if the departure from the entrainment threshold is small enough. With these
assumptions, the mass balance for the tracers on the bed surface reads

$$n_{\mathrm{s}} \frac{\partial \psi}{\partial t} = D\phi - E\psi \tag{3}$$

where $n_s$ is the surface concentration of particles at rest on the bed surface. For homogeneous particles, $n_s \sim 1/d_s^2$ .

For steady and uniform transport, the surface concentration of moving particles, $n$, is constant. In addition, erosion and
deposition balance each other:

$$E = D. \tag{4}$$

Laboratory experiments suggest that the deposition rate is proportional to the concentration of moving particles:

$$D = \frac{n_{\mathrm{m}}}{\tau_f} \tag{5}$$

where we introduce the average flight duration, $\tau_f \propto \ell_f / V$, and $\ell_f$ is the average flight length (Charru et al., 2004; Lajeunesse
et al., 2010b).

Combining equations (2), (3), (4) and (5) provides the set of equations that describe the propagation of the plume:

$$\frac{\partial \phi}{\partial t} + V \frac{\partial \phi}{\partial x} = \frac{1}{\tau_f}(\psi - \phi). \tag{6}$$

$$\frac{\partial \psi}{\partial t} = -\frac{\alpha}{\tau_f}(\psi - \phi) \tag{7}$$

where we define $\alpha = n_{\mathrm{m}}/n_{\mathrm{s}} \sim n_{\mathrm{m}} d_s^2$, the ratio of the concentration of moving particles to the concentration of static particles.
This ratio is smaller than one. It is proportional to the intensity $q_s$ of bedload transport:

$$\alpha \sim \frac{d_s^2}{V} q_s. \tag{8}$$

Complemented with initial and boundary conditions, equations (6) and (7) describe the evolution of the plume. In dimen-
sionless form, they read

$$\frac{\partial \phi}{\partial \hat{t}} + \frac{\partial \phi}{\partial \hat{x}} = \psi - \phi \tag{9}$$

$$\frac{\partial \psi}{\partial \hat{t}} = -\alpha(\psi - \phi). \tag{10}$$





where $\hat{t} = t/\tau_f$ and $\hat{x} = x/\ell_f$ are dimensionless variables. For ease of notation, we drop the hat symbol in what follows.

A single parameter controls equations (9) and (10): the ratio of surface densities $\alpha$, which characterizes the average distance between grains in the bedload layer. Since the erosion-deposition model assumes independent particles, we can only expect it to be valid when moving particles are sufficiently far away from each other, that is when $\alpha$ is small or, equivalently, when the

Shields parameter is near threshold.

In the next section, we solve numerically equations (9) and (10).

## 3   Propagation of a plume of tracers

Laboratory measurements of bedload often use top-view images (Martin et al., 2012; Lajeunesse et al., 2017). Unless individual particles can be tracked, the tracers at rest are usually indistinguishable from those entrained by the flow. Separating the

proportion of tracers in the moving layer, $\phi$, from that on the bed surface, $\psi$, is practically impossible. Instead, top-view pictures show the total concentration of tracers:

$$c = \frac{n_m \phi + n_s \psi}{n_m + n_s} = \frac{\alpha}{\alpha + 1}\phi + \frac{1}{\alpha + 1}\psi. \qquad (11)$$

Tracking sediment in rivers poses a similar problem. In general, one records the position of the tracers when the river stage is below the threshold of grain entrainment (Phillips et al., 2013; Phillips and Jerolmack, 2014). At the time of measurement,

all tracers are therefore at rest. As a result, the proportion of mobile tracers is null, $\phi = 0$, and the total concentration of tracers reads $c = \psi/(\alpha + 1)$.

In summary, the proportions of mobile and static tracers, $\phi$ and $\psi$, naturally derive from mass balance (2) and (3). However their measurement proves difficult during active transport. On the other hand, experimental and field investigations provide the total concentration of tracers, $c$ (Sayre and Hubbell, 1965; Lajeunesse et al., 2017). This quantity is conservative, as the total

amount of tracers, $M = \int c\, dx$, is preserved. In the following, we therefore focus on the concentration of tracers, $c$.

To study the evolution of the tracer concentration, we solve equations (9) and (10) numerically, using a finite volume scheme. We then compute the tracer concentration using equation (11) (Fig. 2).

The early evolution of the plume depends on initial conditions. In most field experiments, tracers are deposited at the surface of the river bed when the flow stage is low and sediment are motionless (Phillips et al., 2013). During floods, the river discharge

increases and the shear stress eventually exceeds the entrainment threshold, setting in motion a small proportion of the grains. The entrainment of particles strongly depends on the arrangement of the bed: grains highly exposed to the flow move first (Charru et al., 2004; Turowski et al., 2011; Agudo and Wierschem, 2012). In most gravel-bed rivers, this configuration is rare. The probability of deposition in a site highly exposed to the flow is therefore small. As a result, tracers rarely move during the first flood. With time however, the bed configuration changes as floods occur repeatedly, and tracers initially trapped in the bed

eventually get entrained by the flow. To simulate this scenario, we assume that, initially, all tracers belong to the static layer:

$$\phi(x,t = 0) = 0.$$

The evolution of the plume follows two distinct regimes. At early times, the flow progressively dislodges tracers from the bed and entrains them in the bedload layer. During this "entrainment regime", only a small proportion of tracers move.



Consequently, the plume develops a thin tail in the downstream direction (Fig. 2a). The corresponding distribution of travel distances is strongly skewed towards the direction of propagation, a feature commonly observed in field experiments (Liébault et al., 2012; Phillips and Jerolmack, 2014).

With time, the plume moves downstream and spreads both upstream and downstream. As a result, the concentration rapidly decreases to small levels. The plume becomes gradually symmetrical and tends asymptotically towards a Gaussian distribution (Fig. 2b). This regime is reminiscent of classical diffusion.

To better illustrate this evolution, we introduce the mean position of the plume of tracers:

$$\langle x \rangle = \frac{1}{M} \int_{-\infty}^{\infty} c\,x\,\mathrm{d}x. \tag{12}$$

We also characterize its size with the variance:

$$\sigma^2 = \frac{1}{M} \int_{-\infty}^{\infty} c\,(x - \langle x \rangle)^2\,\mathrm{d}x \tag{13}$$

and its symmetry with the skewness:

$$\gamma = \frac{1}{\sigma^3} \int_{-\infty}^{\infty} c\,(x - \langle x \rangle)^3\,\mathrm{d}x. \tag{14}$$

The evolution of these three quantities depends on the observation time scale (Fig. 3). During the entrainment regime, the average location of the plume increases as $\langle x \rangle \propto t^2$. The non-linear increase of the variance, $\sigma^2 - \sigma_0^2 \propto t^3$, indicates anomalous diffusion (Nikora et al., 2002; Zhang et al., 2012). In the meantime, its skewness increases as $\gamma \propto t^4$.

After a characteristic time of the order of $\tau \approx \tau_f$, the skewness of the plume reaches a maximum (Fig. 3c). This corresponds to a drastic change of dynamics: the skewness starts decreasing as the plume becomes gradually more symmetrical. At long time, the plume of tracers advances at constant velocity and diffuses linearly with time (Fig. 3a and b). This regime, regardless of the value of $\alpha$, corresponds to classical advection-diffusion.

Next, we establish the equivalence between diffusion and the long-time behavior of the tracers.

## 4 Advection-diffusion at long time

The diffusion at work in equations (9) and (10) results from the continuous exchange of particles between the bedload layer, where particles travel at the constant velocity $V$, and the sediment bed, where particles are at rest. The velocity difference between the two layers gradually smears out the plume and spreads it in the flow direction. This process occurs in a variety of physical systems in which layers moving at different velocities exchange a passive tracer. A typical example is Taylor dispersion, where a passive tracer diffuses across a Poiseuille flow in a circular pipe (Taylor, 1953). The combination of shear rate and transverse molecular diffusion generates an effective diffusion in the flow direction. Other exemples of effective diffusion include solute transport in porous media and chromatography (Van Genuchten and Wierenga, 1976).





To establish formally the equivalence between diffusion and the long-time behavior of the plume, we follow a reasoning similar to the one developed for chromatography (James et al., 2000). Equations (9) and (10) are equivalent to:

$$\frac{\partial c}{\partial t} + \frac{\alpha}{\alpha+1}\frac{\partial c}{\partial x} = \frac{\alpha}{(\alpha+1)^2}\frac{\partial \delta}{\partial x}, \tag{15}$$

$$\frac{\partial \delta}{\partial t} + \frac{1}{\alpha+1}\frac{\partial \delta}{\partial x} + (\alpha+1)\delta = \frac{\partial c}{\partial x}, \tag{16}$$

where we introduce $\delta = \psi - \phi$, the difference between the proportion of tracers on the sediment bed and that in the bedload layer. Eventually, these proportions equilibrate each other. At long time, we therefore expect the solution of equations (15) and (16) to relax towards steady state, for which $\delta$ is of order $\epsilon \ll 1$. Accordingly, we rewrite these two equations as

$$\frac{\partial c}{\partial t} + \frac{\alpha}{\alpha+1}\frac{\partial c}{\partial x} = \epsilon\frac{\alpha}{(\alpha+1)^2}\frac{\partial \delta}{\partial x}, \tag{17}$$

$$\frac{\partial \delta}{\partial t} + \frac{1}{\alpha+1}\frac{\partial \delta}{\partial x} + (\alpha+1)\,\delta = \frac{1}{\epsilon}\frac{\partial c}{\partial x}. \tag{18}$$

Introducing $T = \epsilon\,t$ and $X = \epsilon\,x$, and developing $c$ and $\delta$ with respect to $\epsilon$ yields

$$\frac{\partial c_0}{\partial T} + \frac{\alpha}{\alpha+1}\frac{\partial c_0}{\partial X} = 0 \tag{19}$$

$$(\alpha+1)\,\delta_0 = \frac{\partial c_0}{\partial X} \tag{20}$$

at zeroth order, and

$$\frac{\partial c_1}{\partial T} + \frac{\alpha}{\alpha+1}\frac{\partial c_1}{\partial X} = \frac{\alpha}{(\alpha+1)^2}\frac{\partial \delta_0}{\partial X} \tag{21}$$

at first order.

Multiplying equation (21) by $\epsilon$ and summing the result with equation (19), we finally get

$$\frac{\partial c}{\partial t} + \frac{\alpha}{\alpha+1}\frac{\partial c}{\partial x} = \frac{\alpha}{(\alpha+1)^3}\frac{\partial^2 c}{\partial x^2}. \tag{22}$$

At long time, the transport of the tracers follows the advection-diffusion equation (22) with an advection velocity $U = \alpha/(\alpha + 1) \sim \alpha$ and a diffusion coefficient $D = \alpha/(\alpha + 1)^3 \sim \alpha$. This asymptotic equivalence explains the advection-diffusion regime

in Figures 2 and 3.

In the next section, we investigate the evolution of the location, the size and the symmetry of the plume as it propagates downstream.

## 5   Location, size and symmetry of the plume

Most field campaigns involve a few hundreds of tracer peebles at best (Liébault et al., 2012; Phillips and Jerolmack, 2014). As

the tracers spread over kilometers, their concentration rapidly decreases to immeasurable levels. It is then useful to focus on integral quantities, such as the mean position, $\langle x \rangle$, the size, $\sigma$ or the skewness $\gamma$ of the plume of tracers.



Multiplying equation (15) by $x$ and integrating over space provides the evolution equation for the mean position:

$$\frac{\partial \langle x \rangle}{\partial t} = \frac{\alpha}{\alpha + 1} - \frac{\alpha}{(\alpha+1)^2}\, \langle \delta \rangle \tag{23}$$

where

$$\langle \delta \rangle = \frac{1}{M} \int \delta \, dx \tag{24}$$

is the average difference between the proportion of tracers on the sediment bed and in the bedload layer. To solve equation (23), we need an equation for $\langle \delta \rangle$. The latter is obtained by integrating (16) over space:

$$\frac{\partial \langle \delta \rangle}{\partial t} = -(\alpha + 1)\, \langle \delta \rangle \tag{25}$$

Equations (23) and (25) describe the downstream motion $\langle x \rangle$ of the plume. To solve them, we need to specify initial conditions. As discussed in section 3, we consider that all tracers initially belong to the static layer i.e. $\phi(x, t=0) = 0$. This condition and the conservation of mass, $\langle c \rangle = 1$, provide initial conditions for $\langle \delta \rangle$: $\langle \delta \rangle(t=0) = \alpha + 1$. With this conditions, equations (23) and (25) integrate into

$$\langle\, x\, \rangle - \langle\, x\, \rangle_0 = \frac{\alpha}{\alpha + 1}\, t + \frac{\alpha}{(\alpha+1)^2}\, \left( e^{-(\alpha+1)t} - 1 \right) \tag{26}$$

where $\langle\, x\, \rangle_0$ is the initial position of the plume.

We now focus on the size of the plume. Multiplying (15) by $x^2$ and integrating over space yields the evolution equation for the second moment of the tracer distribution:

$$\frac{\partial \langle x^2 \rangle}{\partial t} = \frac{2\alpha}{(\alpha + 1)}\, \langle x \rangle - \frac{2\alpha}{(\alpha+1)^2}\, \langle x\delta \rangle \tag{27}$$

where

$$\langle x\, \delta \rangle = \frac{1}{M} \int x\, \delta\, dx. \tag{28}$$

is the first moment of $\delta$. To solve equation (27), we need an equation for this intermediate quantity. We obtain it by multiplying (16) by $x$ and integrating over space:

$$\frac{\partial \langle x\delta \rangle}{\partial t} = -1 - (\alpha + 1)\langle x\delta \rangle + \frac{\langle \delta \rangle}{\alpha + 1} \tag{29}$$

At time $t = 0$, $\langle x\delta \rangle(t=0) = (\alpha+1)\langle x \rangle_0$. Equations (27) and (29) with this initial condition provide the expression of the second moment of the tracer distribution:

$$\langle x^2 \rangle = \langle x^2 \rangle_0 + \frac{2\alpha}{(\alpha+1)^3}\, \left( t + \frac{2-\alpha}{\alpha+1} \right) e^{-(\alpha+1)t} + \frac{\alpha^2}{(\alpha+1)^2}\, t^2$$
$$+ \frac{2\alpha\,(1-\alpha)}{(\alpha+1)^3}\, t + \frac{2\alpha\,(\alpha-2)}{(\alpha+1)^4} \tag{30}$$





where $\langle x^2 \rangle_0$ is the initial value of the second moment of the tracer distribution. We then deduce the variance of the plume from:

$$\sigma^2 = \langle x^2 \rangle - \langle x \rangle^2. \tag{31}$$

We follow a similar procedure to derive the skewness of the plume. Multiplying (15) by $x^3$ and integrating over space yields

the evolution equation for the third moment of the tracer distribution:

$$\frac{\partial \langle x^3 \rangle}{\partial t} = \frac{3\alpha}{(\alpha+1)} \langle x^2 \rangle - \frac{3\alpha}{(\alpha+1)^2} \langle x^2 \delta \rangle \tag{32}$$

where

$$\langle x^2 \delta \rangle = \frac{1}{M} \int x^2 \delta \, dx. \tag{33}$$

is the second moment of $\delta$. Multiplying (16) by $x^2$ and integrating over space provides the evolution equation for this interme-

diate quantity:

$$\frac{\partial \langle x^2 \delta \rangle}{\partial t} = -(\alpha+1)\langle x^2 \delta \rangle + \frac{2}{\alpha+1} \langle x \delta \rangle - 2\langle x \rangle \tag{34}$$

At time $t = 0$, $\langle x^2 \delta \rangle = (\alpha+1)\langle x^2 \rangle_0$ and $\langle x^3 \rangle = 0$. With these initial conditions, equations (32) and (34) provide the expression

of $\langle x^3 \rangle$:

$$\langle x^3 \rangle = \frac{3\alpha}{\alpha+1} \left( \sigma_0^2 + \frac{2\alpha^2 - 8\alpha + 2}{(\alpha+1)^4} \right) t$$
$$+ \frac{3\alpha}{(\alpha+1)^2} \left( \sigma_0^2 + \frac{2\alpha^2 - 12\alpha + 6}{(\alpha+1)^4} \right) \left( e^{-(\alpha+1)t} - 1 \right)$$
$$+ \frac{3\alpha}{(\alpha+1)^4} \left( t - 4\frac{\alpha-1}{\alpha+1} \right) t \, e^{-(\alpha+1)t}$$
$$+ \frac{\alpha^3}{(\alpha+1)^3} \left( t - \frac{3(\alpha-2)}{\alpha(\alpha+1)} \right) t^2 \tag{35}$$

from which we deduce the skewness of the plume:

$$\gamma = \frac{\langle x^3 \rangle - 3 \langle x \rangle \, \sigma^2 - \langle x \rangle^3}{\sigma^3} \tag{36}$$

Equations (26), (31) and (36) describe the evolution of the mean, the variance and the skewness of the tracers distribution, that

is, its migration, spreading and symmetry. They do not require any assumption and agree exactly with numerical simulations

(Fig. 3).

As discussed in section 3, numerical simulations reveal a transient during which the tracers, initially at rest, are progressively

set into motion by the flow (Fig. 3). During this entrainment regime, the plume continuously accelerates, spreads non-linearly

and becomes increasingly asymmetrical. To characterize this regime, we expend equations (26), (31) and (36) to leading order



in time:

$$\langle x \rangle - \langle x \rangle_0 \quad \sim \quad \frac{\alpha}{2}\, t^2 \tag{37}$$

$$\sigma^2 - \sigma_0^2 \quad \sim \quad \frac{\alpha}{3}\, t^3 \tag{38}$$

$$\gamma \quad \sim \quad \frac{\alpha}{4\, \sigma_0^3}\, t^4. \tag{39}$$

These three equations are consistent with our numerical simulations (Fig. 3). During the entrainment regime, the variance increases with an exponent larger than one. This behavior is consistent with super-diffusion (Schumer et al., 2009; Ganti et al., 2010; Bradley et al., 2010).

With time, the plume enters the diffusive regime. Its velocity and its spreading rate relax towards constants while its skewness decreases (Fig. 3). We derive the corresponding asymptotic behavior by expending equations (26), (31) and (36) at long time:

$$\langle x \rangle - \langle x \rangle_0 \quad \sim \quad \frac{\alpha}{\alpha + 1}\, t \sim \alpha\, t \tag{40}$$

$$\sigma^2 - \sigma_0^2 \quad \sim \quad 2\frac{\alpha}{(\alpha + 1)^3}\, t \sim 2\, \alpha\, t \tag{41}$$

$$\gamma \quad \sim \frac{3}{\sqrt{2\, \alpha}}\, \frac{1}{\sqrt{t}} \tag{42}$$

The asymptotic regimes (40) and (41) are consistent with the expressions derived in section 4.

The transition between the entrainment and the diffusive regime occurs when the skewness reaches its maximum value. Equating the skewness estimated from (39) and (42) provides the approximate duration of the entrainment regime, $\tau$. We find

$$\tau_e = (72)^{1/9} \left( \frac{\sigma_0^2}{\alpha} \right)^{1/3} \tau_f \tag{43}$$

which compares well with our numerical simulations (Fig. 3). The duration of the entrainement regime increases with the initial size plume and decreases with the intensity of sediment transport.

The asymptotic regimes (37), (38), (39), (40), (41) and (42) assume that sediment transport is in steady state. In the next section, we discuss the intermittency of bedload transport in natural streams.

## 6 Intermittency of bedload transport

Our description of the plume of tracers is based on the assumption that sediment transport is in steady state. This hypothesis is often satisfied in laboratory flumes (Lajeunesse et al., 2017). In a river, it may be met for up to a few days (Sayre and Hubbell, 1965). At longer time scales, however, most rivers alternate between low-flow stages during which sediment is immobile, and floods, during which bed particles propagate downstream (Phillips and Jerolmack, 2016). Bedload transport is thus intermittent.

The intermittency of bedload transport influences the propagation of tracers in several ways. First of all, sediment transport during a flood modifies the structure of the bed (Lenzi et al., 2004; Turowski et al., 2009, 2011). As a result, the proportion of tracers in the bedload layer and in the bed, $\phi$ and $\psi$, likely change from one flood to the next. In a effort to address this question, P. Allemand and collaborators recently implemented the survey of a river located in Basse-Terre Island (Guadeloupe




archipelago). Their preliminary observations reveal that the peebles deposited at the end of a flood are the first entrained at the beginning of the next (P. Allemand, personal communication, June 30, 2017). Based on this observation, we speculate that a tracer belonging to the bedload layer at the end of a flood will still be part of the bedload layer at the beginning of the next one. Similarly, a tracer locked in the bed at the end of a flood will belong to the static layer at the beginning of the next one.

In other words, we assume that tracers freeze between two floods. The plume of tracers therefore keeps the memory of initial conditions, despite floods and low flow stages.

If this assumption holds, the simplest way to account for bedload intermittency is to assume that the river alternates between two representative stages: 1) a low-flow stage during which tracers are immobile; 2) a flood stage, characterized by a representative sediment flux $q_s \sim \alpha V / d_s^2$, during which tracers propagate downstream (Paola et al., 1992). Following this model, we

may extrapolate our results to the field, provided we rescale time with respect to an intermittency factor $I = T_e / T$, where $T$ is the total duration of elapsed time, while $T_e$ is the time during which sediments are effectively in motion (Paola et al., 1992; Parker et al., 1998).

In practice, evaluating the intermittency factor requires continuous monitoring of the river discharge, and a correct estimate of the entrainment threshold. Liébault et al. (2012), for instance, monitored the location of tracer pebbles deposited in the

Bouinenc stream (France) during 2 years. Over this period, the motion of the tracers resulted from 55 floods, for a total duration of 42 days. Sediments were thus in motion less than $I = 12\%$ of the time.

Here, we suggest another way to circumvent the intermittency of sediment transport. Plotting the plume variance, $(\sigma^2 - \sigma_0^2)$, and its skewness, $\gamma$, as a function of traveled distance, $\langle x \rangle - \langle x \rangle_0$, eliminates time from the equations (Fig. 4). In this plot, the position of the plume acts as a proxy for the effective duration of sediment transport, $T_e$. The resulting curves are thus filtered

from transport intermittency (Fig. 4).

The entrainment regime corresponds to small traveled distances. In this regime, both the size of the plume and its asymmetry increase with traveled distance (Fig. 4). Equations (37), (38) and (39) describe the early evolution of the plume. Eliminating time by combining them, we find the behavior of the plume for short traveled distances:

$$\sigma^2 - \sigma_0^2 = \sqrt{\frac{8 \, \ell_f}{9 \, \alpha}} \, (\langle x \rangle - \langle x \rangle_0)^{3/2}, \tag{44}$$


$$\gamma = \frac{\ell_f}{\alpha \, \sigma_0^3} \, (\langle x \rangle - \langle x \rangle_0)^2. \tag{45}$$

As discussed in section 5, the increase of the skewness and the non-linear evolution of the variance indicate anomalous diffusion, due to the progressive entrainment of tracers trapped in the bed.

After the plume has traveled over a distance roughly equal to the flight length, its skewness reaches a maximum value and

starts decreasing. This change of dynamics indicates the transition towards the diffusive regime. Equations (40), (41) and (42) provide the long-term behavior of the plume :

$$\sigma^2 - \sigma_0^2 \sim 2 \, \ell_f \, (\langle x \rangle - \langle x \rangle_0), \tag{46}$$





$$\gamma = \frac{3}{\sqrt{2}} \sqrt{\frac{\ell_f}{\langle x \rangle - \langle x \rangle_0}}. \tag{47}$$

The linear increase of the variance with the distance traveled by the plume is the signature of standard diffusion (see section 5).

Equating the skewness estimated from (45) and (47) provides the distance at which the skewness reaches its maximum:

$$x_{max} \sim \left(\frac{3\,\alpha}{\sqrt{2}}\right)^{2/5} \left(\frac{\sigma_0^6}{\ell_f}\right)^{1/5}. \tag{48}$$

Roughly, the entrainment regime holds as long as the size of the plume is much smaller than the length over which it has traveled.

When expressed in terms of the distance traveled by the plume, the asymptotic regimes are insensitive to the intermittency of bedload transport. As a result, they provide a strong qualitative test of our model. If the validity of the latter extends to field situations, the variance and skewness of the plume plotted as a function of the traveled distance should exhibit the same asymptotic behaviors as in Figure 4.

In this case, the asymptotic regimes may also be used to extract quantitative information from field data. We now illustrate this finding with an example. Suppose that a dataset records the evolution of a plume of tracers released in a river, over a distance long enough to explore both the entrainment and the diffusive regime. During the diffusive regime, the skewness decreases with the traveled distance. In this regime, a fit of the data with equation (47) yields the flight length, $\ell_f$. Knowing the latter, we use equation (45) to deduce the intensity of sediment transport, $\alpha$, from the evolution of the skewness during the entrainment regime.

According to section 5, the skewness reaches a maximum value after a duration $\tau_e$, given by equation (43). Taking into account the intermittency of bedload transport in natural streams, we expect that the maximum is reached when $t = (72)^{1/9}\,(\sigma_0^2/\alpha)^{1/3}\,\tau_f\,/\,I$, where $I$ is the intermittency factor. Field measurement of the time at which the skewness of the plume of tracers reaches its maximum value thus yields the ratio $\tau_f/I$. Combining the latter with our estimates of the flight length, $\ell_f$, and the intensity of sediment transport, $\alpha$, provides the average sediment transport rate in the river:

$$\bar{q}_s = I\,\alpha\,d_s^2\,\frac{l_f}{\tau_f}. \tag{49}$$

## 7 Conclusion

We used the erosion-deposition model introduced by Charru et al. (2004) to describe the evolution of a plume of bedload tracers entrained by a steady flow. In this model, the propagation of the plume results from the stochastic exchange of particles between the bed and the bedload layer. This mechanism is reminiscent of the propagation of tracers in porous media (Berkowitz and Scher, 1998). The evolution of the plume depends on two control parameters: its initial size, $\sigma_0$, and the intensity of sediment transport, $\alpha$.



Our model captures in a single theoretical framework the transition between two asymptotic regimes : 1) an early entrainment regime marked by anomalous diffusion of the plume, 2) a late-time relaxation towards classical advection-diffusion. Both regimes are consistent with previous observations (Nikora et al., 2002; Zhang et al., 2012).

When expressed in terms of the distance traveled by the plume, the asymptotic regimes are insensitive to the intermittency of bedload transport in natural streams. According to this model, it should be possible to estimate the particle flight length and the average bedload transport rate from the evolution of the skewness of a plume of tracers.

*Acknowledgements.* It is a pleasure to thank Pascal Allemand, David Furbish, Colin Phillips, Douglas Jerolmack and François Métivier for many helpfull and enjoyable discussions. This work was supported by the French national programme EC2CO-Biohefect/Ecodyn//Dril/MicrobiEn, *Dispersion de contaminants solides dans le lit d'une rivière*.



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





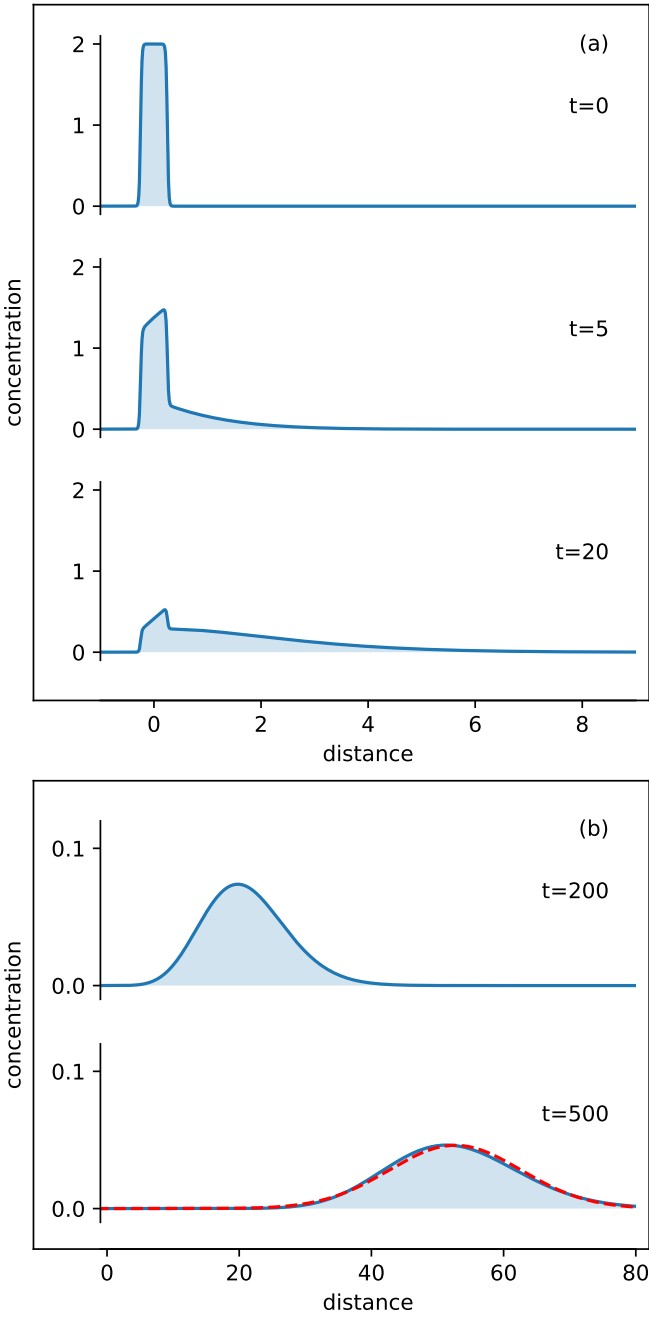

**Figure 2.** Evolution of the tracer concentration ($\alpha = 0.1$) obtained by solving numerically equations (9) and (10). (a) Early entrainment regime. (b) Relaxation towards the diffusive regime. Tracers are initially at rest, forming a symmetric plume of length $L = 0.5$ and mass $M = 1$. The concentration profile asymptotically tends towards a Gaussian distribution (dotted red line).





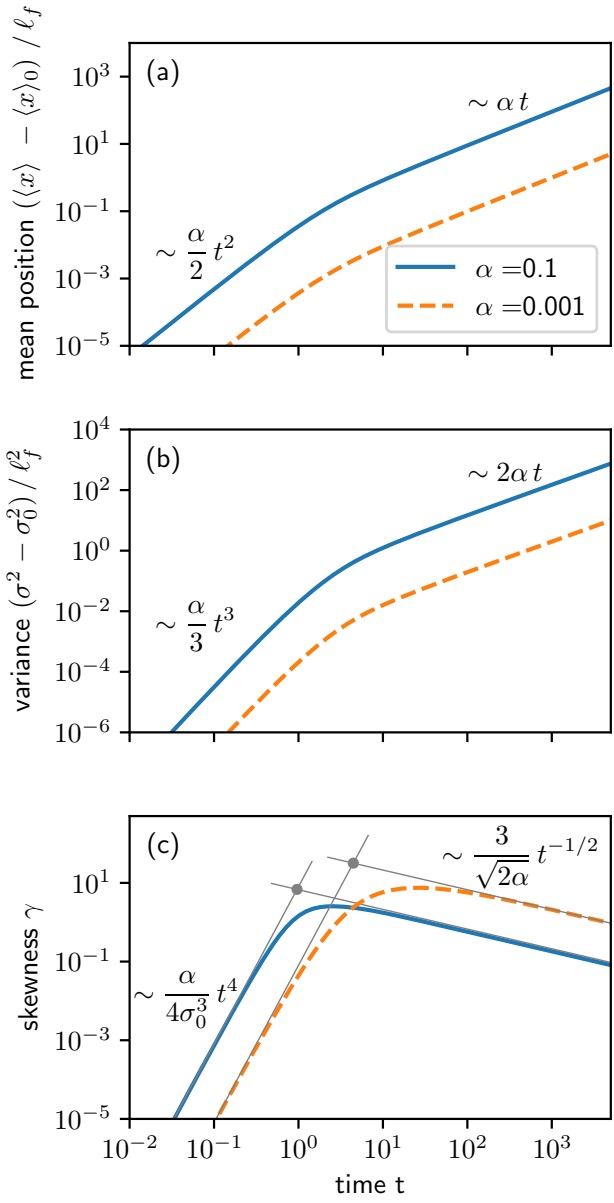

**Figure 3.** (a) Position, (b) variance and (c) skewness of a plume of tracers as a function of time for $\alpha = 0.1$ and $\alpha = 0.001$. We compute the evolution of these three quantities using equations (26), (31) and (36). The results agree exactly with numerical simulations. The asymptotic regimes of the skewness are represented with grey lines. Their intersection provides an estimate of the duration of the entrainment regime (see equation (43)).



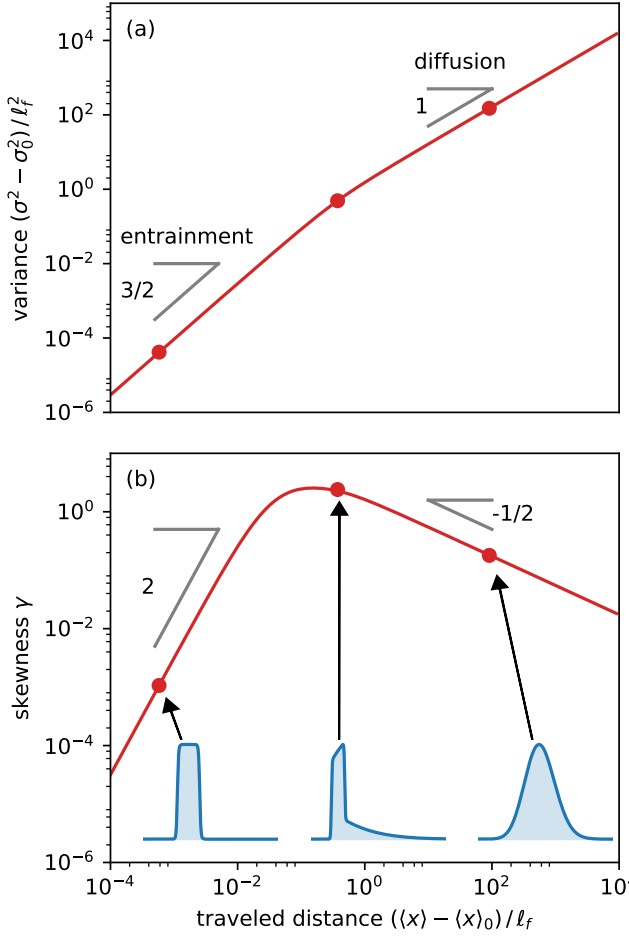

**Figure 4.** (a) Variance and (b) skewness of a plume of tracers as a function of traveled distance ($\alpha = 0.1$). These three quantities are calculated from equations (26), (31) and (36). Inset: concentration profiles (blue) illustrating the shape of the plume during the entrainment regime, at the transition between the entrainment and the diffusive regime and in the diffusive regime.