# Peer review of "Advection and dispersion of bedload tracers"

_Earth Surface Dynamics, 2017_

## Referee Comment (RC1) · Anonymous Referee #1 · 9 Jan 2018

**Review of "Advection and dispersion of bedload tracers" by Eric Lajeunesse, Olivier Devauchelle, and François James**

This paper expands on a previously published model for describing fluvial bedload tracer transport. The model is derived from a mass balance based primarily on the erosion and deposition rates and the fractions of tracer in the mobile and immobile parts of the bed. The model assumes a homogenous bed grain size and steady and uniform transport. From the model, the authors derive analytic expression for the velocity and dispersion of a tracer plume. The paper is well organized and the derivations are mostly clear, with the exception of the derivation of the scaling of the mean and variance of the tracer plume with time. My difficulty following this derivation is perhaps the origin of my most serious concern, discussed below.

My most serious concern is the prediction of how the mean and variance scale with time t during the entrainment regime, specifically that the mean increases as $t^2$ and the variance as $t^3$ (page 6, figure 3, and equations 37-39).

I am not familiar with any literature that describes variance scaling greater than the ballistic regime, variance ~ $t^2$. The *Zhang et al.* [2012] and *Nikora et al.* [2002] references cited on page 6, line 15 do not support this variance scaling. *Weeks and Swinney* [1998] Figure 1 shows an upper limit to the variance scaling of $t^2$ and Table 1 limits the scaling of the mean to $t^x$, x<2. All the observations of anomalous super-diffusion with which I am familiar [e.g. *Bradley*, 2017; *Phillips et al.*, 2013] show a variance scaling of $t^x$ with $1 < x < 2$. The paper needs more discussion of how these predictions are consistent with previous theoretical work and observations of tracer dispersion.

A secondary concern is the connection of the model to the classical advection diffusion equation.

First, it is unsurprising that the model is equivalent to the ADE because the assumptions in the model do not allow for anomalous dispersion that arises from heavy-tailed step lengths or resting times. For example, the assumption of steady and uniform transport without storage in the bed or bars pre-supposes this outcome and limits the model's applicability to real rivers.

Second, I don't see how the advection velocity U and the diffusion coefficient D are related to alpha. Alpha is the ratio of mobile concentration to stationary concentration and it is dimensionless. It's not clear how the authors get from a dimensionless quantity to U with dimensions of L/T and D with $L^2$/T.

My final concern is the use of the mean travel distance as a proxy for transport time as a way to account for the intermittency of transport. While this somehow reduces the variance scaling during the entrainment regime to what *Weeks and Swinney* [1998] define as the allowable range, equating travel distance with time implies a steady tracer virtual velocity and therefore a linear increase of mean tracer position with time. This appears to be inconsistent with the predicted increase in mean position with time (eq. 37) during the entrainment regime. Further discussion of this apparent inconsistency and justification of travel distance as proxy for time is warranted.

**Specific Comments**

The author states on page 2, line 14 that at long times, tracer dispersion is normal, with linearly increasing variance as if it were settled science. In my opinion, this is not a settled issue. Recently published work [*Bradley*] presents evidence of anomalous super-diffusion over 9 years of observation.

Page 4, line 11. The connection between surface grain size concentration and grain size needs clarification. This seems to imply exactly 1 tracer per unit area, an unrealistic assumption.

Page 5, line 15. Phi = 0 is not the same as phi is null. I assume that the authors meant "nil." Null means undefined and not equal to anything. You can never state x = null.

Page 5, line 28-29. The statement that tracers rarely move during the first flood is incorrect and is inconsistent with the statement about tracer installation on the bed surface at the beginning of this paragraph. Nearly all tracer studies neglect the first episode of transport precisely because tracers placed on the bed surface are unnaturally mobile until they are thoroughly mixed into the bed. Similarly, the statement on line 33 that only a small proportion of tracers move during the entrainment regime needs justification. See *Bradley and Tucker* [2012] for example. In the first flood of that study, the proportion of mobile tracers was higher than in a subsequent, nearly identical flood.

Page 7, line 24. It is misleading to say that most tracer studies are limited to a few hundred particles. *Bradley and Tucker* [2012] used nearly 900 tracers.

Page 7. Line 25. The only way that statement that tracer concentration rapidly decreases to immeasurable levels could be correct is if no tracers were recovered. By definition, the recovery of even a single tracer particle is a measurable concentration.

**Technical Corrections**

The word "pebbles" is misspelled as "peebles" in several locations (e.g. page 7, line 24)

Page 2, Line 10 should read "propagate downstream"

Page 2, Line 11: The [*Bradley and Tucker*, 2012] reference is incorrectly cited as Nathan Bradley and Tucker.

Page 9, line 22, should read "expand equations" and the reference to equation 31 is probably intended to be eq. 30.

Page 15, line 20. This reference is incorrectly formatted.

References

Bradley, D. N. (2017), Direct observation of heavy-tailed storage times of bedload tracer particles causing anomalous super-diffusion, Geophysical Research Letters, doi: 10.1002/2017GL075045.

Bradley, D. N., and G. E. Tucker (2012), Measuring gravel transport and dispersion in a mountain river using passive radio tracers, Earth Surface Processes and Landforms, 37(10), 1034-1045, doi: 10.1002/esp.3223.

Nikora, V., H. Habersack, T. Huber, and I. McEwan (2002), On bed particle diffusion in gravel bed flows under weak bed load transport, Water Resour. Res., 38(6), 17.11-19.

Phillips, C. B., R. L. Martin, and D. J. Jerolmack (2013), Impulse framework for unsteady flows reveals superdiffusive bed load transport, Geophysical Research Letters, 40(7), 1328-1333, doi: 10.1002/grl.50323.

Weeks, E., and H. Swinney (1998), Anomalous diffusion resulting from strongly asymmetric random walks, Physical Review E, 57(5), 4915-4920.

Zhang, Y., M. M. Meerschaert, and A. I. Packman (2012), Linking fluvial bed sediment transport across scales, Geophysical Research Letters, 39(20).

---

## Referee Comment (RC2) · Anonymous Referee #2 · 24 Jan 2018

Summary: This manuscript develops an analytical model for the spreading of a plume of bed-load tracers. From the Erosion and Deposition model developed by Charru et al. (2004) they further develop analytical solutions for the mean, variance, and skewness of the spreading tracer plume. This model demonstrates and analytical solutions demonstrate that the spreading of bed-load tracers occupies two scaling regimes. The manuscript further demonstrates that the first three moments of the tracer plume can be set against each other to effectively remove the their dependence on time. They conclude with a useful description of how these results may be tested within a field setting.

General comments: Determining time in a river can be a somewhat abstract exercise and multiple authors have attempted it with varying degrees of success. This difficulty

greatly impairs the utility of field tracers by requiring researchers to monitor both the hydrology and the sediment tracers themselves. However, this manuscript may have provided a framework that greatly increases the utility of field tracers. As the key insight of this manuscript results from setting the expressions for the mean, variance, and skewness against each other and effectively removing time from the problem. This is very clever and to my knowledge has not been done before despite its apparent simplicity (in many ways it would not have made any sense to compare these without the model and framework presented in this manuscript). This framework, if shown to be a reasonable predictor of natural rivers, could take tracers from something of a novelty measurement to a standard tool in bed load and mountain river monitoring campaigns.

I think that this manuscript does a good job of presenting the theory and model development, and I appreciate the authors discussion on how this result can be tested using tracer data as it is rare in the field of sediment transport that theory papers present easy to test hypotheses. These results will likely be of great interest to the bed load transport and mountain river scientific communities.

I have very few comments and they are related primarily to improving the clarity of several variable definitions. The manuscript would benefit from providing a physical description or picture of the flight length and flight duration. From Lajeunesse et al. (2010) these quantities represent the distance a particle travels from erosion to deposition and the duration of this movement, respectively. Those definitions are akin to the descriptions of 'steps' from the many papers that treat bed load probabilistically. In this manuscript though they seem to represent quantities that are much more akin to length and timescales that particles spend on the surface. Making this distinction very clear at the outset would help reader comprehension. Even if these quantities do not quite have an observed definition in the field it would help if the authors could expand on what they think they represent.

In conclusion, I recommend that the manuscript be published in ESurf with a few very minor changes focused on enhancing the clarity.

[In the spirit of ESurf's open discussion period I have elected to read Reviewer 1's comments after the completion of my own review - I did not see anything within Reviewer 1's comments that should prevent this manuscript from being published, however the authors will need to provide greater clarification of their derivations to avoid the issues pointed out by reviewer 1.

A few comments on field tracers and what has been previously observed. To my knowledge all current field datasets report different relations for both the mean and variance scalings, but this is not surprising as these studies all use different metrics for time in a river (some variation of cumulative hydrologic forcing) and the fitted relations almost always stem from regression. Some of these regressions are physically justified, but the main point here is that a lot of different relations could be fit to the available datasets. That no one has really observed multiple mean and variance scaling regimes is not surprising. Without apriori knowledge of multiple scaling regimes and the locations of the break points it is unlikely that one would ever try to fit a complex function to these data due to the variability. With this current paper, there is no a reason to attempt more complicated models for the field data.

A final comment on the length of observation in field studies and a contribution that this manuscript makes. Even for the longest observed field studies (9 yrs as pointed out by Reviewer 1) it is not clear how long the rivers in those studies are actually 'on' (actively able to transport sediment). In a sense, a decade in a desert stream with few floods could be the same as a month in a tropical river that floods weekly. In terms of dynamics, maybe 9 yrs of data represents the entire scaling regime and maybe it still only represents the entrainment regime, because most of the time gravel rivers are effectively 'off'. This is key result of the current manuscript, as it provides a way to compare tracer studies by removing time, one of the more nebulous variables.]

Specific comments:

In several locations the term 'pebble' is used in place of what are likely cobbles. I

understand what the authors mean, however more traditional geologists may find the use of the term confusing and misinterpret the size of the particles in question. I leave it up to the authors to choose.

Description of equation 1 - It is not immediately clear what the unit surface is? Is this the projected area of a grain ($D^2$) or the measurement window?

P. 4 Ln. 18 - The introduction of the 'flight length' should include a definition. Although it is defined in the cited papers, a short definition would benefit the readers comprehension of the concept. Something like the flight length represents the distance a particle travels from erosion to deposition.

P. 6 Ln. 16 - It now becomes clear to me that I am not sure exactly what tau_f (the flight time) refers to physically. Is it the time of an individual flight (from erosion to deposition in the surface layer, on the order of seconds) or does it refer to a longer timescale that represents the time that the particle remains in a more mobile state?

P. 7 Ln. 24 & P. 11 Ln. 1 - 'peebles' likely a typo for pebbles. Though I would suggest cobbles per the earlier comment. Bradley and Tucker (2012) or Bradley (2017) would be worth citing here as it represents the largest deployment to date.

P. 7 Ln. 26 - is 'the size' supposed to be the standard deviation?

P. 8 Ln. 10 - 'this' should be 'these' if the conditions are indeed plural.

P. 11 Ln. 6 - The preceding lines set up the notion that tracers maintain their conditions between floods (they don't move and in a sense are frozen), but this line suggests that this also applies to the actual floods. It is just a little confusing, during the flood isn't that when tracers might be mobile and thus changing their conditions? Please clarify what is meant in this line and if floods should be included.

Pg. 11 Ln. 10-12 - Based on Paola et al. (1992), Phillips et al. (2013) have partially validated that the hydrograph intermittency is proportional to this same quantity. You might cite them here as a validation for the frameworks potentially broad applicability.

Pg. 12 Ln. 7 - It is not immediately clear to me what this line is saying. Could you reword this sentence to clarify its meaning. What I gathered from it is that the plume of tracers will remain in the entrainment scaling regime so long as the size (variance or range?) is less than the length (mean?) position. Is this what is meant?

References: Bradley, D. N., and G. E. Tucker (2012), Measuring gravel transport and dispersion in a mountain river using passive radio tracers, Earth Surface Processes and Landforms, 37(10), 1034-1045, doi: 10.1002/esp.3223.

Bradley, D. N. (2017), Direct observation of heavy-tailed storage times of bedload tracer particles causing anomalous super-diffusion, Geophysical Research Letters, doi: 10.1002/2017GL075045.

---

## Author Comment (AC1) · 5 Mar 2018

We thank the referees for their careful reading of our paper, and for the issues they raised. They will find below our answers, and the changes made to the text. For the sake of clarify, we reproduce the referees's comments in italics, followed by our answers. Similarly, the changes applied to the article appear in blue on the new manuscript, uploaded as supplementary material.

[Figure]

**1 Response to referee 1**

**1.1 General comments**

*This paper expands on a previously published model for describing fluvial bedload tracer transport. The model is derived from a mass balance based primarily on the erosion and deposition rates and the fractions of tracer in the mobile and immobile parts of the bed. The model assumes a homogenous bed grain size and steady and uniform transport. From the model, the authors derive analytic expression for the velocity and dispersion of a tracer plume. The paper is well organized and the derivations are mostly clear, with the exception of the derivation of the scaling of the mean and variance of the tracer plume with time. My difficulty following this derivation is perhaps the origin of my most serious concern, discussed below.*

*a My most serious concern is the prediction of how the mean and variance scale with time t during the entrainment regime, specifically that the mean increases as $t^2$ and the variance as $t^3$ (page 6, figure 3, and equations 37-39). I am not familiar with any literature that describes variance scaling greater than the ballistic regime, variance $\sim t^2$. The Zhang et al. [2012] and Nikora et al. [2002] references cited on page 6, line 15 do not support this variance scaling. Weeks and Swinney [1998] Figure 1 shows an upper limit to the variance scaling of $t^2$ and Table 1 limits the scaling of the mean to $t^x, x < 2$. All the observations of anomalous super-diffusion with which I am familiar [e.g. Bradley, 2017; Phillips et al., 2013] show a variance scaling of $t^x$ with $1 < x < 2$. The paper needs more discussion of how these predictions are consistent with previous theoretical work and observations of tracer dispersion.*

As the referee points out, super-diffusion cannot lead to variance increasing faster than $\sim t^2$. Our claim that the entrainment regime involves super-diffusion was incorrect. In fact, the scalings derived for the entrainment regime show that it is of a different nature

than super-diffusion. We have changed several paragraphs in sections 3, 5, 6, and in the conclusion: page 7- lines 30-33, and page 15 - line 26. The new manuscript also includes the following discussion, stressing out the difference between anomalous diffusion and the entrainment regime (page 12 lines 21 - page 13 line 7):

"Anomalous diffusion arises from heavy-tailed distributions, of either the step length or the waiting time (Weeks and Swinney, 1998). The erosion-deposition model contains no such ingredient. Here the fast increase of the variance results from the exchange of particles between the sediment bed and the bedload layer, at the beginning of the experiment. Over a time shorter than the flight duration $\tau_f$, the tracers entrained by the flow do not settle back on the bed. They form a thin tail, which leaves the main body of the plume, and moves downstream at the average particle velocity $V$ (Fig. 2 a). The plume therefore consists of a main body of virtually constant concentration, followed by a thin tail of length $\propto Vt$. Accordingly, we can split the integral that defines its mean position, equation (12), into two terms. The first one, obtained by integrating $cx$ over the main body of the plume, yields the initial position of the plume $\langle x \rangle_0$. The second one, obtained by integrating $cx$ over a tail of length $Vt$, scales as $t^2$. Summing these contributions yields equation (39). Similar reasonings yield equations (40) and (41) for the variance and the skewness."

**b** *A secondary concern is the connection of the model to the classical advection diffusion equation.*

*First, it is unsurprising that the model is equivalent to the ADE because the assumptions in the model do not allow for anomalous dispersion that arises from heavy-tailed step lengths or resting times. For example, the assumption of steady and uniform*

*transport without storage in the bed or bars pre- supposes this outcome and limits the model's applicability to real rivers.*

As discussed above, the erosion-deposition model contains none of the ingredients necessary for super-diffusion. Following the referee's comment, we have included in the new manuscript a discussion about the physical mechanism at work in the advection-diffusion regime (page 10 - lines 9-13):

"We interpret this formal derivation as follows. In the reference frame of the plume, a tracer at rest on the bed moves backward, while a tracer entrained in the bedload layer moves forward. At long time, the proportions of tracers in each layer equilibrate. Consequently, the probability that a tracer be entrained and move forward, equals that of deposition. In the reference frame of the plume, the exchange of particles between the bed and the bedload layer is thus a Brownian motion, hence the linear diffusion of the plume."

*Second, I don't see how the advection velocity U and the diffusion coefficient D are related to $\alpha$. $\alpha$ is the ratio of mobile concentration to stationary concentration and it is dimensionless. It's not clear how the authors get from a dimensionless quantity to U with dimensions of L/T and D with L2/T.*

As of section 3, all equations are dimensionless. Accordingly, the expression of the advection velocity U and the diffusion coefficient D derived at the end of section 4 are dimensionless. With dimensions, they read:

$$U = \frac{\alpha}{\alpha+1} \frac{\ell_f}{\tau_f} \tag{1}$$

$$D = \frac{\alpha}{(\alpha+1)^3} \frac{\ell_f^2}{\tau_f} \tag{2}$$

We have edited the corresponding equations at the end of section 4 to clarify this point.

*My final concern is the use of the mean travel distance as a proxy for transport time as a way to account for the intermittency of transport. While this somehow reduces the variance scaling during the entrainment regime to what Weeks and Swinney [1998] define as the allowable range, equating travel distance with time implies a steady tracer virtual velocity and therefore a linear increase of mean tracer position with time. This appears to be inconsistent with the predicted increase in mean position with time (eq. 37) during the entrainment regime. Further discussion of this apparent inconsistency and justification of travel distance as proxy for time is warranted.*

We dot not equate travel distance with time. As the referee points out, this would be inconsistent with equation (37) (and several others). Instead, we note that setting the expressions for the mean, variance, and skewness against each other removes time from the problem. Combining equations (37), (38) and (39) even provides analytical expression of the variance and skewness as a function of traveled distance during the entrainment regime. However, interpreting the resulting scaling as a signature of anomalous diffusion was wrong. As discussed above, the entrainment regime is not a super-diffusive regime. We thank the referee for pointing out this inconsistency. We changed section 6 accordingly.

**1.2 Specific comments**

*The author states on page 2, line 14 that at long times, tracer dispersion is normal, with linearly increasing variance as if it were settled science. In my opinion, this is not a settled issue. Recently published work [Bradley] presents evidence of anomalous super-diffusion over 9 years of observation.*

We changed the corresponding paragraph (page 2, lines 11-25):

> "The dispersion of the tracers, expressed as the variance of their location, results from the randomness of bedload transport. Nikora *et al.* (2002) identify three regimes with distinct time scales. A particle entrained by the flow repeatedly collides with the bed (Lajeunesse *et al.*, 2017). At short time, between two collisions, particles move with the flow, and the variance increases as the square of time (Martin, Jerolmack, and Schumer, 2012; Fathel, Furbish, and Schmeeckle, 2016). This regime is analogous to the ballistic regime of Brownian motion (Zhang, Meerschaert, and Packman, 2012; Fathel, Furbish, and Schmeeckle, 2016).
>
> As the particle continues its course, collisions deviate its trajectory. In this intermediate regime, the variance increases non-linearly with time (Martin, Jerolmack, and Schumer, 2012). Nikora *et al.* (2002) attribute this behavior to anomalous super-diffusion; but Fathel, Furbish, and Schmeeckle (2016) contest their interpretation.
>
> With time, tracers settle back on the bed, where they can remain trapped for a long time. How the distribution of resting times influences the long-term dispersion of tracers remains unknown. The data collected by Sayre and Hubbell (1965) are consistent with the existence of a diffusive regime, in which the variance increases linearly (Zhang, Meerschaert, and Packman, 2012). Other investigators, however, report either subdiffusion or super-diffusion (Nikora *et al.*, 2002; Bradley, 2017). These anomalous diffusion regimes are sometimes modeled with fractional advection-dispersion equations (Schumer, Meerschaert, and Baeumer, 2009; Ganti *et al.*, 2010; Bradley, Tucker, and Benson, 2010). "

*Page 4, line 11. The connection between surface grain size concentration and grain size needs clarification. This seems to imply exactly 1 tracer per unit area, an unrealistic assumption.*

The sediment bed is made of uniform particles of size $d_s$. Each of them occupies an area which scales like $\sim d_s^2$. The surface concentration of particles at rest on the bed surface is therefore $n_s \sim 1/d_s^2$. The concentration of moving particles is much smaller, $n \sim \alpha/d_s^2$. We have edited the corresponding paragraph to clarify this point (page 4, lines 9-10).

*Page 5, line 15. Phi = 0 is not the same as phi is null. I assume that the authors meant "nil." Null means undefined and not equal to anything. You can never state x = null.*

This sentence is now : "As a result, the proportion of mobile tracers vanishes ($\phi = 0$), and the total concentration of tracers reads $c = \psi/(\alpha + 1)$."

*Page 5, line 28-29. The statement that tracers rarely move during the first flood is incorrect and is inconsistent with the statement about tracer installation on the bed surface at the beginning of this paragraph. Nearly all tracer studies neglect the first episode of transport precisely because tracers placed on the bed surface are unnaturally mobile until they are thoroughly mixed into the bed.*

We have edited the corresponding paragraph to clarify this point (page 6 - lines 7-15) :

"The early evolution of the plume depends on initial conditions. In most field experiments, tracers are deposited at the surface of the river bed when the flow stage is low and sediment are motionless (Phillips, Martin, and Jerolmack, 2013). During floods, the river discharge increases and the

shear stress eventually exceeds the entrainment threshold, setting in motion some of the grains. The entrainment of particles strongly depends on the arrangement of the bed: grains highly exposed to the flow move first (Charru, Mouilleron, and Eiff, 2004; Turowski, Badoux, and Rickenmann, 2011; Agudo and Wierschem, 2012). Several authors find that the tracers they disposed on the bed are more mobile during the first flood than during later ones (Bradley and Tucker, 2012). During the later floods, tracers gradually get trapped in the bed, and their average mobility decreases. On the other hand, Phillips and Jerolmack (2014) find no special mobility during the first flood. In the absence of a clear scenario, we choose the simplest possible initial conditions: we assume that, initially, all tracers belong to the static layer: $\phi(x, t = 0) = 0$."

*Similarly, the statement on line 33 that only a small proportion of tracers move during the entrainment regime needs justification. See Bradley and Tucker [2012] for example. In the first flood of that study, the proportion of mobile tracers was higher than in a subsequent, nearly identical flood.*

The statement on line 33 is not about field studies. It describes the behavior of the plume predicted from equations (9) and (10), subject to the initial condition $\phi(x, t = 0) = 0$. We have modified the corresponding paragraph to clarify this point (page 6, line 16) :

"With these initial conditions, the evolution of the plume follows two distinct regimes. At early times, the flow gradually dislodges tracers from the bed and entrains them in the bedload layer. During this entrainment regime, only a small proportion of the tracers move."

*Page 7, line 24. It is misleading to say that most tracer studies are limited to a few hundred particles. Bradley and Tucker [2012] used nearly 900 tracers. Page 7. Line 25. The only way that statement that tracer concentration rapidly decreases to immeasurable levels could be correct is if no tracers were recovered. By definition, the recovery of even a single tracer particle is a measurable concentration.*

These two statements (Page 7, line 24 and 25) are not meant as a criticism of field measurements. Instead, they question the use of the concentration for comparison between theory and field data. We have reworked the corresponding paragraph to clarify these two points (page 10, lines 17-24):

> "Concentration, defined as the number of tracers per unit of area, depends on the area over which it is measured. Its value is meaningful when the measurement area is much larger than the distance between particles, and much smaller than the plume. During the entrainment regime, the plume develops a thin tail containing only a small proportion of tracers. Measuring the concentration profile during this regime is thus challenging. To our knowledge, only Sayre and Hubbell (1965) were able to measure consistent concentration profiles, using radioactive sand. In practice, most field campaigns involve a limited number of tracers (900 at most) (Liébault *et al.*, 2012; Bradley and Tucker, 2012; Phillips and Jerolmack, 2014; Bradley, 2017). It is thus more practical to consider integral quantities, such as the mean position of the plume $\langle x \rangle$, its variance $\sigma^2$, and its skewness $\gamma$."

**1.3 Technical corrections**

*The word "pebbles" is misspelled as "peebles" in several locations (e.g. page 7, line 24)*

Following the recommandation of reviewer #2, we have changed pebble into cobble.

*Page 2, Line 10 should read "propagate downstream"*

It now reads "travel downstream".

*Page 2, Line 11: The [Bradley and Tucker, 2012] reference is incorrectly cited as Nathan Bradley and Tucker.*

We corrected the reference.

*Page 9, line 22, should read "expand equations" and the reference to equation 31 is probably intended to be eq. 30.*

We corrected the spelling of expand. Regarding the equations, (31) is the correct reference. But the development needs equation (30).

*Page 15, line 20. This reference is incorrectly formatted.*

We corrected the reference.

**2   Response to referee 2**

**2.1   General comments**

*Summary: This manuscript develops an analytical model for the spreading of a plume of bed-load tracers. From the Erosion and Deposition model developed by Charru et al. (2004) they further develop analytical solutions for the mean, variance, and skewness of the spreading tracer plume. This model demonstrates and analytical solutions demonstrate that the spreading of bed-load tracers occupies two scaling regimes. The manuscript further demonstrates that the first three moments of the tracer plume can be set against each other to effectively remove the their dependence on time. They conclude with a useful description of how these results may be tested within a field setting.*

*General comments: Determining time in a river can be a somewhat abstract exercise and multiple authors have attempted it with varying degrees of success. This difficulty greatly impairs the utility of field tracers by requiring researchers to monitor both the hydrology and the sediment tracers themselves. However, this manuscript may have provided a framework that greatly increases the utility of field tracers. As the key insight of this manuscript results from setting the expressions for the mean, variance, and skewness against each other and effectively removing time from the problem. This is very clever and to my knowledge has not been done before despite its apparent simplicity (in many ways it would not have made any sense to compare these without the model and framework presented in this manuscript). This framework, if shown to be a reasonable predictor of natural rivers, could take tracers from something of a novelty measurement to a standard tool in bed load and mountain river monitoring campaigns.*

*I think that this manuscript does a good job of presenting the theory and model development, and I appreciate the authors discussion on how this result can be tested using tracer data as it is rare in the field of sediment transport that theory papers present*

*easy to test hypotheses. These results will likely be of great interest to the bed load transport and mountain river scientific communities.*

***a** I have very few comments and they are related primarily to improving the clarity of several variable definitions. The manuscript would benefit from providing a physical description or picture of the flight length and flight duration. From Lajeunesse et al. (2010) these quantities represent the distance a particle travels from erosion to deposition and the duration of this movement, respectively. Those definitions are akin to the descriptions of 'steps' from the many papers that treat bed load probabilistically. In this manuscript though they seem to represent quantities that are much more akin to length and timescales that particles spend on the surface. Making this distinction very clear at the outset would help reader comprehension. Even if these quantities do not quite have an observed definition in the field it would help if the authors could expand on what they think they represent.*

As pointed out by the referee, the definition of the flight length and duration were ambiguous. We have modified the text following equation (5), to clarify these definitions in section 2 (page 4, line 17 - page 5, line 2):

> "Laboratory experiments suggest that the deposition rate is proportional to the concentration of moving particles:
>
> $$D = \frac{n_{\mathrm{m}}}{\tau_f} \tag{3}$$
>
> where we introduce the average flight duration, $\tau_f = \ell_f / V$, and the average flight length, $\ell_f$ (Charru, Mouilleron, and Eiff, 2004; Lajeunesse, Malverti, and Charru, 2010). The flight length is the distance traveled by a mobile particle between its erosion and eventual deposition. Similarly, the flight duration is the time a particle spends in the bedload layer. In practice, measuring these quantities often proves difficult, since they depend on how one defines the mobile and the static layer (Lajeunesse *et al.*, 2017)."

*In conclusion, I recommend that the manuscript be published in ESurf with a few very minor changes focused on enhancing the clarity.*

*[In the spirit of ESurf's open discussion period I have elected to read Reviewer 1's comments after the completion of my own review - I did not see anything within Reviewer 1's comments that should prevent this manuscript from being published, however the authors will need to provide greater clarification of their derivations to avoid the issues pointed out by reviewer 1.*

*A few comments on field tracers and what has been previously observed. To my knowledge all current field datasets report different relations for both the mean and variance scalings, but this is not surprising as these studies all use different metrics for time in a river (some variation of cumulative hydrologic forcing) and the fitted relations almost always stem from regression. Some of these regressions are physically justified, but the main point here is that a lot of different relations could be fit to the available datasets. That no one has really observed multiple mean and variance scaling regimes is not surprising. Without apriori knowledge of multiple scaling regimes and the locations of the break points it is unlikely that one would ever try to fit a complex function to these data due to the variability. With this current paper, there is no a reason to attempt more complicated models for the field data.*

*A final comment on the length of observation in field studies and a contribution that this manuscript makes. Even for the longest observed field studies (9 yrs as pointed out by Reviewer 1) it is not clear how long the rivers in those studies are actually 'on' (actively able to transport sediment). In a sense, a decade in a desert stream with few floods could be the same as a month in a tropical river that floods weekly. In terms of dynamics, maybe 9 yrs of data represents the entire scaling regime and maybe it still only represents the entrainment regime, because most of the time gravel rivers are effectively 'off'. This is key result of the current manuscript, as it provides a way to compare tracer studies by removing time, one of the more nebulous variables.*

[Figure]

**2.2   Specific comments**

*In several locations the term 'pebble' is used in place of what are likely cobbles. I understand what the authors mean, however more traditional geologists may find the use of the term confusing and misinterpret the size of the particles in question. I leave it up to the authors to choose.*

Following the reviewer's advice, we have replace the term 'pebble' with 'cobble' everywhere in the mansucript.

*Description of equation 1 - It is not immediately clear what the unit surface is? Is this the projected area of a grain (DĔĘ2) or the measurement window?*

We are not referring to some specific surface, but to a unit of area, i.e. $1\ m^2$. $E$ is the number of bed particles set in motion per unit of time and area. We have edited the text to clarify this ambiguity (page 3 - lines 26-27).

*P. 4 Ln. 18 - The introduction of the 'flight length' should include a definition. Although it is defined in the cited papers, a short definition would benefit the readers comprehension of the concept. Something like the flight length represents the distance a particle travels from erosion to deposition.*
*P. 6 Ln. 16 - It now becomes clear to me that I am not sure exactly what $tau_f$ (the flight time) refers to physically. Is it the time of an individual flight (from erosion to deposition in the surface layer, on the order of seconds) or does it refer to a longer timescale that represents the time that the particle remains in a more mobile state?*

As discussed above, we have edited the text to clarify the definitions of the flight length and duration (page 4, line 17 - page 5, line 2).

*P. 7 Ln. 24 & P. 11 Ln. 1 - 'peebles' likely a typo for pebbles. Though I would suggest cobbles per the earlier comment. Bradley and Tucker (2012) or Bradley (2017) would be worth citing here as it represents the largest deployment to date.*

As discussed above, we have substituded 'pebble' for 'cobble' everywhere in the manuscript. Following referee # 1, the first paragraph of section 5 has been reworked. It now includes citations of Bradley and Tucker (2012) and Bradley (2017).

*P. 7 Ln. 26 - is 'the size' supposed to be the standard deviation?*

Indeed, the size is the standard deviation.To clarify this ambiguity, we have changed the text into " It is thus more practical to consider integral quantities, such as the mean position of the plume $\langle x \rangle$, its variance $\sigma^2$, and its skewness $\gamma$." (page 10, line 23).

*P. 8 Ln. 10 - 'this' should be 'these' if the conditions are indeed plural.*

Actually, there is only one condition : $\langle \delta \rangle$: $\langle \delta \rangle (t = 0) = \alpha + 1$.

*P. 11 Ln. 6 - The preceding lines set up the notion that tracers maintain their conditions between floods (they don't move and in a sense are frozen), but this line suggests that this also applies to the actual floods. It is just a little confusing, during the flood isn't that when tracers might be mobile and thus changing their conditions? Please clarify what is meant in this line and if floods should be included.*

We removed this sentence, which was both confusing and unnecessary.

*Pg. 11 Ln. 10-12 - Based on Paola et al. (1992), Phillips et al. (2013) have partially validated that the hydrograph intermittency is proportional to this same quantity. You might cite them here as a validation for the frameworks potentially broad applicability.*
We have added a reference to Phillips, Martin, and Jerolmack (2013).

*Pg. 12 Ln. 7 - It is not immediately clear to me what this line is saying. Could you reword this sentence to clarify its meaning. What I gathered from it is that the plume of tracers will remain in the entrainment scaling regime so long as the size (variance or range?) is less than the length (mean?) position. Is this what is meant?*

This sentence was not only unclear, it was also incorrect. We changed it to (page 16, lines 6): "The entrainment regime lasts until the plume has traveled over a distance comparable to its initial size, that is until $\langle x \rangle - \langle x \rangle_0 \sim \sigma_0$."

**ESurfD**

**Supplement:**

[revised manuscript text omitted]